# The experiences and needs of unpaid family caregivers for persons living with dementia in rural settings: A qualitative systematic review

Heather J. Campbell-Enns[1]*, Stephen Bornstein[2,3], Veronica M. M. Hutchings[4], Maya Janzen[5‡], Melanie Kampen[6‡], Kelli O'Brien[7,8], Kendra L. Rieger[9], Tara Stewart[10], Benjamin Rich Zendel[3,4], Malcolm B. Doupe[10]

1 Department of Psychology, Canadian Mennonite University, Winnipeg, Manitoba, Canada, 2 Department of Political Science, Memorial University, St. John's, Newfoundland, Canada, 3 Faculty of Medicine, Memorial University, St. John's, Newfoundland, Canada, 4 Aging Research Centre-NL, Grenfell Campus, Memorial University, Corner Brook, Newfoundland and Labrador, St. John's, Canada, 5 School of Public Policy and Administration, Carleton University, Ottawa, Ontario, Canada, 6 Centre for Transnational Mennonite Studies, University of Winnipeg, Winnipeg, Manitoba, Canada, 7 St. Joseph's Care Group, Thunder Bay, Ontario, Canada, 8 Western Health, Corner Brook, Newfoundland and Labrador, Canada, 9 School of Nursing, Trinity Western University, Langley, British Columbia, Canada, 10 Department of Community Health Sciences, University of Manitoba, Winnipeg, Manitoba, Canada

☯ These authors contributed equally to this work.
‡ These authors also contributed equally to this work.
* hcampbell-enns@cmu.ca

**Data Availability Statement:** All relevant data are within the paper and its Supporting Information files.

## Abstract

### Purpose

Unpaid family caregivers provide extensive support for community-dwelling persons living with dementia, impacting family caregivers' health and wellbeing. Further, unpaid family caregiving in rural settings has additional challenges because of lower access to services. This systematic review examines qualitative evidence to summarize the experiences and needs of rural unpaid family caregivers of persons living with dementia.

### Methods

CINAHL, SCOPUS, EMBASE, Web of Science, PsychINFO, ProQuest, and Medline were searched for articles investigating the experience and needs of rural family caregivers of persons living with dementia. Eligibility criteria were: 1) original qualitative research; 2) written in the English language; 3) focused on the perspectives of caregivers of community-dwelling persons with dementia; 4) focused on rural settings. Study findings were extracted from each article and a meta-aggregate process was used to synthesize the findings.

### Findings

Of the 510 articles screened, 36 studies were included in this review. Studies were of moderate to high quality and produced 245 findings that were analyzed to produce three synthesized findings: 1) the challenge of dementia care; 2) rural limitations; 3) rural opportunities.

**Funding:** This research was supported by a grant awarded to H.C.E. from the Canadian Institutes of Health Research (Funding Reference Number #163082). www.cihr-irsc.gc.ca The funders had no role in study design, data collection and analysis, decision to publish, or preparation of the manuscript.

**Competing interests:** The authors have declared that no completing interests exist.

## Conclusions

Rurality is perceived as a limitation for family caregivers in relation to the scope of services provided but can be perceived as a benefit when caregivers experience trustworthy and helpful social networks in rural settings. Implications for practice include establishing and empowering community groups to partner in the provision of care. Further research must be conducted to better understand the strengths and limitations of rurality on caregiving.

## Introduction

Dementia is an umbrella term for diseases characterized by the progressive deterioration of cognitive abilities and functional ability [1] that impact approximately 55 million people worldwide [2]. With 10 million new diagnoses every year, the global prevalence of dementia is projected to be 139 million people in 2050 [2]. While dementia is a key cause of dependency and disability for older adults [3], many persons with dementia live at home in community settings [4]. In Canada, for example, 61% of older persons with a dementia diagnosis live outside of institutional facilities (i.e., long-term care) [5]. While many of these individuals have mild or moderate dementias and low-care needs, some have complex care needs which can result in heavy caregiver burden. Canadian data show that 20% of community-dwelling persons with dementia have severe cognitive impairment, 25% exhibit responsive behaviours, 25% show signs of depression, and 28% require extensive assistance or are completely dependent on others for activities of daily living tasks like dressing and eating [5].

Unpaid family caregivers provide extensive support to community-dwelling people living with dementia. For example, they provide essential tasks such as supervision for persons with dementia and assistance with their activities of daily living. This care would account for over 40% of the total cost of dementia care if family caregivers were financially compensated for their work [6]. While most people living with dementia prefer to live at home, many are transferred to institutions when the care burden becomes too great for family caregivers [7].

While unpaid family caregivers recognize and report positive impacts of their caregiving [8], the demands of this care can negatively impact them by decreasing their overall psychological wellbeing and overall health [9], increasing health care use [9], and increasing chronic stress [4], social isolation and depression [10]. A recent systematic review [11] of family caregiver needs highlighted that they have specific needs (e.g., information, formal and informal support needs) related to the management of persons with dementia and also related to their own personal lives (e.g., helping to manage their own physical and psychological health) [11].

It is important to note that the geographical location where persons living with dementia and their caregivers reside influences their experiences and needs [12]. While definitions of rurality vary, most are based on population size or distance from services [12,13]. While rural unpaid family caregivers provide more hours of caregiving per week than their urban counterparts [14], it is unclear whether this is due to the inaccessibility or limited availability of formal services, the lack of other family and/or social support in rural settings, isolation exacerbated by geography and/or demographics, or some combination of these factors [12,15]. While family caregiver burden has been reported as being no different in rural areas than in urban areas, this may be because rural family caregivers have adapted to a lack of services by relying on informal supports [16]. Previous reviews on unpaid family caregiving in rural areas have not focussed on dementia care [13] or the qualitative experience of these caregivers [13,17].

New knowledge is needed to address gaps in dementia care literature, including the paucity of information on the experiences of family caregivers in rural communities, as well as their

supportive care needs. Understanding how rurality influences family caregiving is necessary so family caregivers can be better supported, and family members can, in turn, better support persons living with dementia to live in their community settings as long as possible.

Accordingly, a systematic review was undertaken to address these gaps. A qualitative review was required to synthesize data across previous studies which represent the experiences of family caregivers in their own words. Thus, we conducted a systematic review of qualitative studies to summarize the experiences and needs of unpaid family caregivers living in rural settings while caring for persons living with dementia. Our overall question was "What are the experiences and needs of unpaid family caregivers of persons living with dementia in rural settings"? This review is intended to increase understanding of the family caregiver experience for health system leaders and health care providers who distribute resources to rural areas, as well as for organizations providing dementia education to families and communities. We also aim for this review to encourage further research to contribute to an expansive understanding of the experiences and needs of family caregivers in rural settings.

## Methods

A review protocol was registered and published with the PROSPERO database (ID: CRD42020163912). The PICo (population, phenomenon of interest, context) framework [18] was used to guide the development of this qualitative review. Accordingly, we focused on to the population of interest (family caregivers for persons living with dementia), the phenomenon of interest (experience and needs in relation to caregiver burden and stress and interactions with informal and formal care systems), and the context (rural setting). The Joanna Briggs Institute (JBI) method for qualitative systematic reviews was followed and a meta-aggregation method was used to collect and analyze the data [19]. Meta-aggregation is rooted in pragmatism and aims to examine qualitative evidence (e.g., experiences and meanings) and produce results in the form of generalizable statements applicable to policy or practice [19]. According to Lockwood et al. (2015), meta-aggregative reviews feature a defined objective or question, detailed inclusion and exclusion criteria, a comprehensive search strategy, quality appraisal of the included studies, analysis of data extracted, presentation and synthesis of findings, and transparency in the approach taken.

All authors participated in designing this review. The search strategy was designed in consultation with a university librarian. Three authors conducted the literature search, article selection, quality appraisal, data extraction, and synthesis of the data. An initial review took place from December 2019 to September 2020, but writing the review was paused due to the slowing of research activities during the COVID-19 pandemic. As a result, a second search was conducted in Feb 2022 to capture and add new literature. All authors provided feedback on the synthesis and were involved in writing the resulting report.

### Eligibility criteria

To be included, studies must have been original qualitative research written in English and focused on the experiences and needs of unpaid family caregivers of persons living with dementia in rural communities. Mixed-methods studies with a qualitative component were accepted if the qualitative data could be extracted for this review. Several terms were broadly defined to capture as many pertinent studies as possible: 1) "family" was defined as anyone a person living with dementia would consider to be family; 2) "caregiver" was defined as an unpaid family carer; 3) "dementia" captured both Alzheimer's Disease and any other type of dementia; and, 4) "rural" included studies using the terms rural or remote to describe their

settings. As well, studies were included with representatives of both rural and urban caregivers only in cases where the rural caregiver perspective could be extracted from the article.

Studies were excluded from this review if they did not: 1) focus on the experiences or needs of unpaid family caregivers of community-dwelling persons living with dementia; 2) include quotations from caregivers to support their findings; or, 3) focus on rural settings. Articles were also excluded if: 4) the full paper could not be found; and 5) the paper was not original research.

## Search

The following search strategy was developed in consultation with a university librarian, and the following terms were used consistently across the online databases: (dementia OR alzheimer* OR aging) AND (rural OR remote) AND (care* OR caregiver burden) AND (interview OR qualitative). Online databases were searched from their inception to December 2021, including CINAHL, SCOPUS, EMBASE, Web of Science, PsychINFO, ProQuest Dissertations & Theses, and MEDLINE. Google Scholar and Open Access Theses and Dissertations and other relevant websites were searched, as well as the reference lists of relevant articles and reviews.

## Article selection

All titles and abstracts were screened by two independent reviewers to determine if they met the inclusion criteria. Full text versions of potentially relevant studies were retrieved and independently assessed by two reviewers to confirm their eligibility. Disagreements were resolved through discussion and, in the case of disagreement or uncertainty on article inclusion, a third reviewer was consulted. All eligible articles were assessed using the JBI Critical Appraisal Tool for Qualitative Research.

## Data extraction

Authors developed a data extraction form to extract study details. The form was tested on three initial study articles and then revised as needed. Two authors extracted data independently and consolidated the data through discussion. If there was uncertainty, a third reviewer was involved. The form included study location, year of publication, purpose as stated by authors, methods, sample size, participant characteristics, setting, specific study findings, and an illustrative participant quotation for each study finding. The findings from each study were located and documented by examining the results sections of the included studies and were not derived through re-interpretation by this review.

## Data analysis and synthesis

Data analysis software (ATLAS.ti) was utilized to support the comparison of the extracted findings (e.g., themes) in the meta-aggregative process. All extracted findings (e.g., themes) and representative quotations were coded independently for their meanings and compared by three authors. Extracted findings were also coded for whether they explicitly referred to experiences or needs related to rural settings. Extracted findings were assembled into categories based on similarity of concept meaning (multiple findings per category) and the categories were then synthesized (with multiple categories being represented in each synthesized finding). These synthesized findings are overall descriptions of a group of categories derived from combining findings and aim to be representative of the evidence being combined [19].

# Results

## Study characteristics

The search strategy (Fig 1) retrieved 704 citations. After duplicates were removed, 510 articles were identified as potentially relevant to the objectives of the review. Titles and abstracts of the articles were examined, and 425 articles were excluded from the review because they did not meet the inclusion criteria. The full text of the remaining 85 articles were reviewed. After a full review, 49 further articles were excluded, and the remaining 36 articles were included in this review (Table 1)

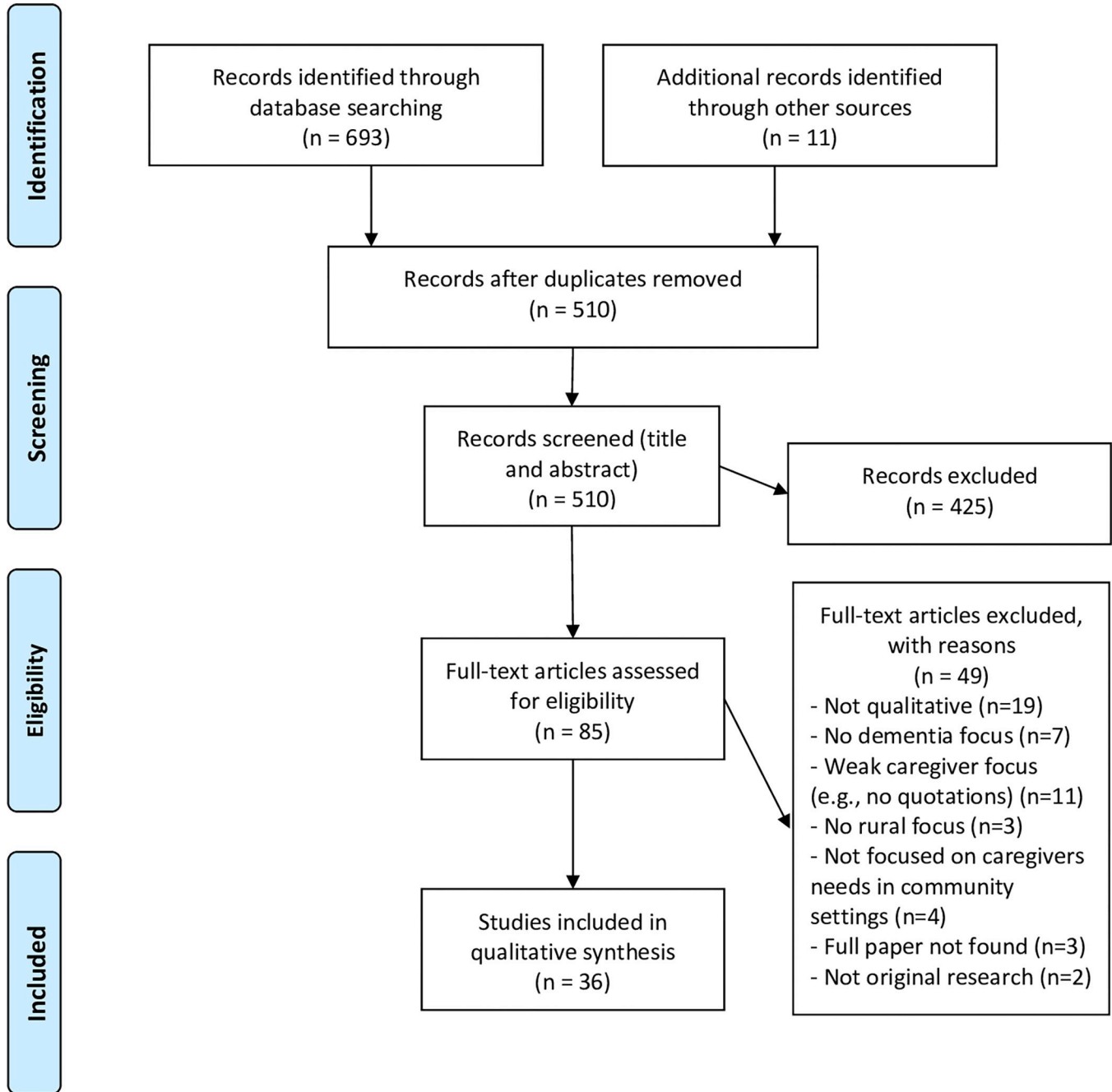

**Fig 1. Flow diagram of research process.**

**Table 1. Included studies and numbered findings.**

| Author(s), year, country | Purpose | Methodology; methods | Family caregivers | Findings | Quality rating |
|---|---|---|---|---|---|
| Agyeman et al. [20], 2019, Ghana | To explore the sociocultural beliefs, understandings, perceptions, and behaviours relating to living with dementia. | Qualitative; semi-structured focus groups; inductive thematic analysis | n = 18; gender/sex data are unclear; age data are unclear | 1. Symptoms<br>2. Understanding of cognitive symptoms<br>3. Help seeking<br>4. Course of cognitive illness<br>5. Care arrangements<br>6. Economic impacts of care<br>7. Stigma | 7/10 |
| Arai et al. [21], 2000, Japan | To determine whether opinions of others may discourage the use of public services for the elderly. | Mixed methods; semi-structured interviews and thematic analysis | n = 7; gender/sex data are unclear; age data are unclear | 8. Fear of not being deemed a dutiful daughter-in-law<br>9. Stigma attached to public services<br>10. Invasion of privacy | 6/10 |
| Blackstock et al. [22], 2006, Scotland | To develop a qualitative understanding of people with dementia and their carers' experiences of using services. | Qualitative; semi-structured interviews, focus groups, field notes and thematic analysis | n = 30; 22 women, 8 men; age range: 40–90 years | 11. Local social networks as informal support services<br>12. Supportive communities: Alternative perspectives<br>13. Personal relationship to place and space<br>14. Qualifying rurality: Implications of time and space | 7/10 |
| Branger et al. [23], 2016, Canada | To describe how rural caregivers cope with caring for a loved one diagnosed with mild cognitive impairment or dementia. | Qualitative description; an open-ended question on a survey and qualitative content analysis | n = 166; gender/sex data are unclear; age data are unclear | 15. Social support: Help with planning and decision-making<br>16. Social support: Professional services<br>17. Time for self: Relaxing activity<br>18. Time for self: Physical activity<br>19. Time for self: Work and career<br>20. Time for self: Routine<br>21. Restructuring: Behavioural changes<br>22. Restructuring: Cognitive changes<br>23. Faith and religion<br>24. Joint or reciprocal activity<br>25. Checking in | 7/10 |
| Cox et al. [24], 2019, Australia | To illustrate (Australian) Aboriginal community understandings of dementia and their responses to dementia care. | Community participatory action research (two-stage mixed methods approach); individual interviews and thematic analysis | n = 12; gender/sex data are unclear; age data are unclear | 26. Community understanding of dementia<br>27. Dementia care as cultural obligation<br>28. Dementia care support | 8/10 |
| Di Gregorio et al. [25], 2015, Canada | To explore the stages of the dementia journey through the viewpoints of health services providers, caregivers, community members, and people living with dementia. | Qualitative; individual interviews, group interviews and thematic analysis | n = 15; gender/sex data are unclear; age data are unclear | 29. Community awareness of dementia<br>30. Receiving a diagnosis<br>31. Progression to long term care | 8/10 |
| Duggleby et al. [26], 2009, Canada | To explore the experience of home for family members caring for a person with dementia. | Grounded theory; Open-ended interviews and constructivist grounded theory approach to analysis | n = 17; 14 females, 3 males; age range: 48–84 years | 32. Social context: The everyday experience<br>33. Fading hope as the main concern<br>34. Renewing everyday hope<br>35. Coming to terms<br>36. Finding positives<br>37. Seeing possibilities | 8/10 |
| Ehrlich et al. [27], 2017, Sweden | To explore and better understand the interrelationship between the caregiving experience of family caregivers and the sociocultural sphere that urban and rural areas represent. | Hermeneutical approach; open-ended interviews and hermeneutical analysis | n = 23; 9 females, 3 males; age range: 48–83 years | 38. Accepting the course of life<br>39. Preserving normalcy<br>40. Fulfilling obligations | 9/10 |
| Forbes & Blake et al. [28], 2013, Canada | To enhance understanding of the process of knowledge sharing among health care practitioners, caregivers, and persons with dementia within a rural First Nations community. | Constructivist grounded theory; interviews, field notes, memos, and grounded theory analysis | n = 3; gender/sex data are unclear; age data are unclear | 41. Developing trusting relationships: Supporting the person with dementia<br>42. Developing trusting relationships: Setting the stage<br>43. Developing trusting relationships: Resolving conflicts<br>44. Accessing and adapting the information: Accessing information during the early stages of dementia<br>45. Accessing and adapting the information: Accessing information during the late stage of dementia<br>46. Accessing and adapting the information: Accessing information from a variety of sources<br>47. Accessing and adapting the information: Inequitable resources in the community<br>48. Applying the information: Applying health promotion strategies<br>49. Involving caregivers in the decision-making | 8/10 |

*(Continued)*

**Table 1.** (Continued)

| Author(s), year, country | Purpose | Methodology; methods | Family caregivers | Findings | Quality rating |
|---|---|---|---|---|---|
| Forbes & Finkelstein et al. [29], 2012, Canada | To examine information needs, how these change over time, and how they access, assess, and apply the knowledge. | Qualitative interpretive description; interviews and thematic analysis | n = 14; 11 females, 3 males; age range: 39–91 years | 50. Recognizing the symptoms 51. Receiving a diagnosis 52. Loss of independence 53. Initiating and using respite programs 54. Long term care facility placement 55. End of life care 56. Barriers to knowledge exchange: Lack of adequate rural community-based services and supports 57. Facilitators of knowledge exchange: Collaborating with team members 58. Expertise: Trusting relationships | 8/10 |
| Forbes & Markle-Reid et al. [30], 2008, Canada | To explore the use and satisfaction with home and community-based service for persons with dementia from the perspectives of caregivers. | Qualitative; focus groups, semi-structured interviews, and thematic analysis | n = 39; 82% females, 18% males; age data are unclear | 59. Availability of home and community-based services 60. Acceptability of home and community-based services: Comprehensive assessments, treatment, and provision of dementia care 61. Acceptability of home and community-based services: Inconsistency of care provider 62. Acceptability of home and community-based services: Attributes of trusting partnerships 63. Acceptability of home and community-based services: Inflexible care 64. Acceptability of home and community-based services: Cost of services | 8/10 |
| Gibson et al. [31], 2019, United States | To conduct an inductive, thematic, analysis focusing on rural caregivers' perceptions of familial and community supports. | Qualitative; semi-structured interviews and thematic analysis | n = 11; 81.8% females, 18.2% males; age range: 57–84 years | 65. Rural caregivers on caregiving role, expectations, and values Helpful resources 66. and supports for rural caregivers 67. Unmet needs and unhelpful resources for rural caregivers | 7/10 |
| Gray-Vickrey [32], 1993, United States | To describe the lived experience of caring for a spouse with Alzheimer's Disease. | Phenomenology; focus groups, semi-structured interviews, and phenomenological analysis | n = 11; 8 females, 3 males; age range: 41–87 years | 68. Caretaking: Obligations and rewards of providing care 69. Caretaking: Caregivers are faced with problem behaviours 70. Caretaking; Emotions regarding caregiving 71. Caretaking: Prisoners in their own homes 72. Caretaking: No time or space to themselves 73. Caretaking: Financial burden 74. Caretaking: Caregivers feel constantly tired 75. Caretaking: Role change 76. Caretaking: Physical and emotional consequences of caregiving 77. Caretaking: Caregiving is time consuming 78. Caretaking: Coping with problem behaviours 79. Caretaking: Distancing 80. Marital relationship: Loss of companionship 81. Marital relationship: Caregiver's distress at watching spouse's decline 82. Changes in social support: Others fail to recognize the caregiver's burden 83. Changes in social support: Caregiver is the other victim 84. Changes in social support: Social isolation 85. Changes in social support: Lack of appropriate or adequate informal support 86. Changes in social support: Caregiver perceptions of satisfactory informal support 87. Changes in social support: Support from support groups 88. Changes in social support: Conflicting feelings about the health care delivery system 89. Changes in social support: Inadequacy of nursing homes 90. Changes in social support: Home and health aide support 91. Understanding the Alzheimer's process: Awareness of the disease process 92. Understanding the Alzheimer's process: Knowledge about Alzheimer's Disease 93. Understanding the Alzheimer's process: Concern about the future | 8/10 |

*(Continued)*

**Table 1.** (Continued)

| Author(s), year, country | Purpose | Methodology; methods | Family caregivers | Findings | Quality rating |
|---|---|---|---|---|---|
| Gurayah [33], 2012, South Africa | An exploratory study into the phenomenon of caregiving for people with dementia in a rural context in South Africa. | Phenomenology; semi-structured interviews and thematic analysis | n = 5; 4 females, 1 male; age range: 46–68 years | 94. Views and responsibilities of the caregiver<br>95. Impact of caregiving<br>96. Services to assist the caregiver | 8/10 |
| Herron & Rosenberg [34], 2017, Canada | Examine how people with dementia experience their communities and support within them. | Qualitative case study approach; semi-structured interviews, field notes and constant comparison approach to analysis | n = 43; gender/sex data are unclear; age data are unclear | 97. Recognizing community connections and contributions<br>98. Negotiating space outside the home<br>99. Identifying care needs and negotiating service use | 7/10 |
| Herron & Rosenberg [35], 2019, Canada | Examine experiences of providing and accessing care over the course of dementia and across settings. | Qualitative case study approach; semi-structured interviews and constant comparison approach to analysis | n = 27; gender/sex data are unclear; age range: 46–89 years | 100. Navigating the system<br>101. Finding people who understand<br>102. Seeking home care hours<br>103. Resistance to respite<br>104. Making decisions about end-of-life care | 7/10 |
| Herron, Rosenberg & Skinner [36], 2016, Canada | To explore how partners in care negotiate support over time and across different settings and the major challenges over the course of the condition. | Mixed methods; in-depth structured interviews and constant comparison approach to analysis | n = 43; gender/sex data are unclear; age data are unclear | 105. Rural experiences of voluntary sector service | 7/10 |
| Herron & Skinner [37], 2013, Canada | To make visible the over-looked emotional experiences of older rural people and their carers, and to reveal the unjust conditions of care within the rural context. | Qualitative; in-depth interviews, focus groups and thematic analysis | n = 14; 10 females, 4 males; age range: 50–80 years | 106. Interpersonal challenges<br>107. Caring places: Feeling at home | 7/10 |
| Hughes et al. [38], 2009, United States | To examine the experiences of African American caregivers that led them to seek a formal diagnosis for a family member. | Qualitative; semi-structured interviews and grounded theory analysis | n = 17; 14 females, 3 males; age range: 42–80 years | 108. Participants' prior knowledge of Alzheimer's disease<br>109. Perceptions of perceived severity of Alzheimer's disease<br>110. Perceptions of susceptibility to Alzheimer's disease<br>111. Perceptions of facilitators and barriers to a diagnosis<br>112. Perceptions of benefits to a diagnosis<br>113. Cues to action<br>114. Perceptions of self-efficacy | 7/10 |
| Innes et al. [39], 2005, Scotland | To develop a qualitative understanding of service use from the point of view of people with dementia and their carers in rural Scotland. | Qualitative; semi-structured interviews, focus groups and use of a coding structure. | n = 30; 22 females, 8 males; age data are unclear | 115. Gaps in services available: Transport<br>116. Gaps in services available: Respite provision<br>117. Gaps in services available: Support for informal carers<br>118. Gaps in services available: Home care<br>119. Gaps in services available: Day care<br>120. Positive experiences of service provision: Services appropriate for needs<br>121. Positive experiences of service provision: Loving care<br>122. Positive experiences of service provision: Social life<br>123. Positive experiences of service provision: Stimulating<br>124. Positive experiences of service provision: Relationship with service providers<br>125. Positive experiences of service provision: Support<br>126. Positive experiences of service provision: Free time<br>127. Positive experiences of service provision: peace of mind<br>128. Positive experiences of service provision: Trained staff<br>129. Positive experiences of service provision: Service provider relationship with carer | 7/10 |

**Table 1.** (Continued)

| Author(s), year, country | Purpose | Methodology; methods | Family caregivers | Findings | Quality rating |
|---|---|---|---|---|---|
| Innes et al. [40], 2014, Scotland | To explore the reported difficulties and satisfactions and diagnostic processes and post-diagnostic support offered to people with dementia and their families living in the largest remote and rural region in Scotland. | Qualitative; semi-structured interviews and thematic analysis | n = 12; 11 females, 1 male; age range: 45–80 years | 130. Pre-diagnosis: Recognising the problem and seeking help<br>131. Pre-diagnosis: Rationalization and denial<br>132. Experience of the diagnostic process<br>133. Experience of the diagnostic process: Delivery of the diagnosis<br>134. Experience of the diagnostic process: Reaction to the diagnosis<br>135. Post-diagnostic support: The needs of service users<br>136. Post-diagnostic support: Satisfaction with services<br>137. Post-diagnostic support: Support for carers<br>138. Post-diagnostic support: The need for information | 7/10 |
| Larsen et al. [41], 2017, Norway | To explore how formal and family caregivers experience collaboration while providing home-based dementia care, with a focus on user participation. | Qualitative; semi-structured interviews and thematic analysis | n = 7; gender/ sex data are unclear; age data are unclear | 139. Negotiating participation in decisions: They no longer know what is best for them<br>140. Negotiating participation in decisions: The person living at home is the boss<br>141. Negotiating participation in decisions: Acute necessary health care<br>142. Negotiating formal care intervention: Family caregivers want to preserve normality<br>Negotiating formal 143. care intervention: Family caregivers' care burden breakdown<br>144. Negotiating the right to speak on behalf: Family caregivers fight for resources<br>145. Negotiating the right to speak on behalf: The most troubling issue | 8/10 |
| Lilly et al. [42], 2012, Canada | To investigate the health and wellness and support needs of family caregivers to persons with dementia in the Canadian policy environment. | Qualitative; focus group and thematic analysis | n = 19; 16 females, 3 males; age range unclear | 146. Forgotten: Abandoned to care, alone and indefinitely<br>147. Information and referral<br>148. Adequate and appropriate in-home service provision for their care recipient<br>149. Respite and relief services for caregivers<br>150. Assistance with care recipients' transitions into long term care<br>151. Unrealistic expectations for caregiver self-care | 7/10 |
| Mattos et al. [43], 2019, United States | To explore perceived social determinants of health among older, rural-dwelling adults with early-stage cognitive impairment. | Qualitative description; semi-structured interviews and thematic analysis | n = 10; 7 females, 3 males; age range: 64–78 years | 152. Staying active<br>153. Eating well<br>154. Living with cognitive changes<br>155. Advantages of living rural<br>156. Disadvantages of living rural<br>157. Relying on children<br>158. Connecting with neighbours and community | 8/10 |
| Morgan et al. [44], 2002, Canada | To describe the community-based process and pilot project used to develop a study of rural dementia care. | Mixed methods; focus group and thematic analysis | n = 4; 4 females; age data are unclear | 159. Stigma of dementia<br>160. Acceptability and accessibility of services<br>161. Service delivery challenges<br>162. Consequences of not using support services | 7/10 |
| Musyimi et al. [54], 2021, Kenya | To explore perceptions and experiences of dementia and related care in rural Kenya | Qualitative; in-depth interviews and focus groups; inductive thematic analysis | n = 12; 8 females, 4 males; age range: 25–86 years | 163. Negative stereotypes of dementia: Negative labels attached to dementia<br>164. Negative stereotypes of dementia: Traditions and cultural beliefs<br>165. Negative stereotypes of dementia: Normal ageing and ageism<br>166. Limited knowledge about dementia care and treatment: Carers' knowledge<br>167. Diagnostic pathway<br>168.Neglect and abuse | 7/10 |
| Nguyen et al. [55], 2021, Vietnam | To describe the meanings of dementia and the day-to-day lived experience of family caregiving in an area just outside of central Hanoi. | Descriptive qualitative; interviews; thematic analysis | n = 12; 6 females, 6 males; age range: 48–62 years | 169. Perceptions of dementia<br>170. Family caregiving as moral obligation<br>171. Gender and birth order in family caregiving<br>172. Difficulties and challenges of family caregiving: Time demands<br>173. Difficulties and challenges of family caregiving: Loss of income<br>174. Difficulties and challenges of family caregiving: Worsening of physical health<br>175. Difficulties and challenges of family caregiving: Increased social isolation<br>176. Difficulties and challenges of family caregiving: Emotional distress | 8/10 |

*(Continued)*

**Table 1.** (Continued)

| Author(s), year, country | Purpose | Methodology; methods | Family caregivers | Findings | Quality rating |
|---|---|---|---|---|---|
| Orpin et al. [45], 2014, Australia | To explore patterns of formal and informal support utilisation by people caring for a person with dementia in a rural-regional context. | Unstated methodology; semi-structured interviews and thematic analysis | n = 18; 8 females, 10 males; age range: 77–83 years | 177. The rural context<br>178. The primary carer role<br>179. Patterns of carer support | 3/10 |
| Prorok et al. [46], 2017, Canada | To examine the perceived primary care health care experiences of both PWD and their caregivers in Ontario, Canada, using qualitative methods. | Qualitative; focus groups and thematic analysis | n = 21; 14 females, 7 males; age range: 45–81 years | 180. Communication: Content communicated<br>181. Communication: Day to day management<br>182. Communication: Long term management<br>183. Communication: Managing self<br>184. System navigation: 'Point person' necessary<br>185. System navigation: Prolonged path to resources and supports<br>186. Ease of access: Timing<br>187. Ease of access: Provider knowledge | 7/10 |
| Sanders [47], 2007, United States | To examine the experience of male caregivers with their informal support networks. | Phenomenology; semi-structured interviews and phenomenological analysis | n = 20; 20 males; age range: 41–84 years | 188. Perception of the willingness of the informal support networks to provide help: Not involved with care<br>189. Perception of the willingness of the informal support networks to provide help: Emergency assistance only<br>190. Perception of the willingness of the informal support networks to provide help: Feel free to call if we could be of help<br>191. Perception of the willingness of the informal support networks to provide help: Part of the caregiving team<br>192. Willingness of the male caregiver to ask for help: Asked for assistance<br>193. Willingness of the male caregiver to ask for help: Felt guilty about asking for help<br>194. Willingness of the male caregiver to ask for help: Did not ask for help | 8/10 |
| Sanders & McFarland et al. [48], 2002, United States | To determine what factors lead sons to assume a primary caregiver role for a parent with progressive memory loss and the caregiving challenges most commonly experienced. | Grounded theory; unstructured interviews and use of coding processes | n = 18; 18 males; age range: 35–67 years | 195. Initial reaction to memory loss<br>196. Becoming the caregiver<br>197. Women in the sons' world<br>198. Learning new roles<br>199. Conflicts<br>200. Uncomfortable situations<br>201. Service utilization | 8/10 |
| Sanders & Power et al. [49], 2009, United States | To examine the changes that occur in the roles, responsibilities, and relationships husbands who provide care for their wives with memory loss and other chronic health conditions. | Phenomenology; semi-structured interviews and phenomenological analysis | n = 17; 17 males; age range: 66–85 years | 202. Adaptation of old roles to new roles due to increased responsibility: Protector of self-esteem, dignity, and personhood<br>203. Adaptation of old roles to new roles due to increased responsibility: Provider of personal care<br>204. Adaptation of old roles to new roles due to increased responsibility: Planner of activities and socialization<br>205. Adaptation of old roles to new roles due to increased responsibility: Home maintenance and keeper<br>206. Developing new relationships with their wives: Developing a new type of intimacy and closeness<br>207. Developing new relationships with their wives: Adjusting to the personality changes associated with chronic illness<br>208. Developing new relationships with their wives: Learning to cope with the unexpected in their relationship<br>209. Developing new relationships with their wives: Recognizing the finality of the relationship | 8/10 |

(*Continued*)

**Table 1.** (Continued)

| Author(s), year, country | Purpose | Methodology; methods | Family caregivers | Findings | Quality rating |
|---|---|---|---|---|---|
| Smith, A. et al. [50], 2001, United States | To learn more about the process involved in living as a primary caregiver of an Alzheimer's patient. In addition, the goal was to learn more about the needs of caregivers in the struggles they faced. | Qualitative; semi-structured interviews and analysis methods were unclear | n = 45; 39 females, 6 males; age data are unclear | 210. Financial assistance<br>211. Legal assistance<br>212. Medical assistance<br>213. Housing assistance<br>214. Emotional support/assistance<br>215. Spirituality | 5/10 |
| Smith, K. et al. [51], 2011, Australia | To determine ways to overcome factors affecting the successful delivery of services to Aboriginal people with dementia living in remote communities, and to their families and communities. | Qualitative; in-depth interviews and thematic analysis | n = 31; gender/sex data are unclear; age data are unclear | 216. Caregiver role: Reasons for being a caregiver<br>217. Caregiver role: additional care giving responsibilities<br>218. Caregiver role: Sharing care-giving role<br>219. Perspectives of dementia: Causes of dementia<br>220. Perspectives of dementia: Signs and symptoms<br>221. Community and culturally appropriate care: Community engagement<br>222. Community and culturally appropriate care: Community-based care<br>223. Community and culturally appropriate care: Culturally appropriate activities<br>224. Workforce: Aboriginal staff<br>225. Workforce: Staff that are trusted and accepted<br>226. Workforce: Local support and guidance<br>227. Education and training: Dementia training<br>228. Issues affecting remote communities: Transport<br>229. Service issues: 229. Communication and coordination<br>230.Services issues: Caregiver support services | 7/10 |
| Vellone et al. [52], 2012, Canada | To explore the meaning of quality of life for Sardinian caregivers of people affected with Alzheimer's disease and factors improving and worsening their quality of life. | Phenomenology; open-ended interviews and extraction of themes | n = 41; 35 females, 6 males; age range: 26–78 years | 231. What is quality of life? Unity and cooperation in the family<br>232. What is quality of life: Freedom, independence, having time for themselves<br>233. What is quality of life? Serenity/tranquility<br>234. What is quality of life? Well-being and health<br>235. Factors worsening quality of life–fear for the future: For the care needed and for the illness and worsening<br>236. Factors worsening quality of life–continuous care of the patients: not having time for themselves<br>237. Factors improving quality of life: No worsening of the illness<br>238. Factors improving quality of life: Help and support from family<br>239. Factors improving quality of life: Help from formal services<br>240. Factors improving quality of life: Satisfaction and reward from giving care<br>241. Factors improving quality of life: Financial support for paying for assistance<br>242. Factors improving quality of life: More free time<br>243. Factors improving quality of life: More public sensitization about Alzheimer's disease | 9/10 |
| Wiersma & Denton [53], 2016, Canada | To explore and understand the context of dementia in rural northern communities in Ontario with an emphasis on understanding how dementia friendly the communities were. | Qualitative interpretivist constructionist paradigm; in-depth interviews, field notes and coding processes for analysis | n = 15; gender/sex data are unclear; age data are unclear | 244. Looking out for the person with dementia<br>245. Remaining connected | 8/10 |

[20–53]. These 36 studies represented a total of 847 unpaid family caregiver participants (ranging from 3 to 166 participants). Thirty-four studies focused solely on rural caregivers and two studies included both rural and urban caregivers [32,38]. As reported in Table 1, studies were conducted by authors who were affiliated with institutions in Canada (n = 15), the United States (n = 7), Australia (n = 3), Scotland (n = 3), Norway (n = 1), Sweden (n = 1), South Africa (n = 1), Kenya (n = 1), Ghana (n = 1), and Japan (n = 1).

### Study quality

The quality appraisal checklist consisted of 10 items, and, except for one low-quality article, all the studies were of moderate to good quality overall. No article included all 10 items (average = 7.1 of 10; range = 3 to 9 of 10) (Table 1). The most frequently excluded quality items were a statement locating the researcher culturally or theoretically (n = 35), the influence of the researcher on the research and vice-versa (n = 32), and a stated philosophical perspective and research methodology (n = 21).

We also report on whether and how rurality was defined. In half of the studies, the authors did not objectively define this term [21,24,26,28,30,31,33,38,41,46–52,54,55]. In the remaining studies, authors defined rurality by various criteria–geographic location [36,39,40], proximity to urban centres or distance to services [23,29,34,37,43,45], population density [27,42], by the agricultural or resource-based economy of the area [20,35], or the perceived rural identity of participants [34,36]. As well, several studies used a combination of the above factors in defining rurality; various studies utilized a combination of population density and proximity to urban centres [22,25,32,53], a combination of population density and type of economy [44], or a combination of geographic location, economic factors, and perceived rural identity [36].

### Synthesized findings

Data extraction resulted in 245 study findings from the 36 included articles. Extracted findings were numbered to track their contributions to the synthesized findings (Table 1). After coding each extracted finding, codes were aggregated into 11 categories. These categories were 1) coping, 2) stress and emotions, 3) finances, 4) role/identity of caregiver, 5) noticing health changes, 6) family support, 7) getting help and/or adapting, 8) communication, 9) decision-making, 10) services provided and, 11) information and understanding. Then, through a process of reading and re-reading category contents, categories were then synthesized into three synthesized findings to describe the unpaid family caregiver experience, with two findings particular to the rural setting. Table 2 shows how categories contributed to synthesized findings by referring to extracted finding numbers. No extracted findings were omitted from the synthesis. The three synthesized findings are described below.

### Synthesized finding 1: The challenge posed by dementia for rural caregivers

"The challenge posed by dementia" describes the difficult or demanding experiences related to being an unpaid family caregiver of a person with dementia, as perceived by participants living in rural areas. As participants did not explicitly link many of these experiences with providing unpaid care in rural settings, it is unknown if these experiences are related to, or exacerbated by, the rural context. Since this synthesized finding captures the overall experience of unpaid family caregiving, all 11 categories (Table 2) contributed to this synthesized finding.

First, unpaid family caregivers reported that seeking a dementia diagnosis was challenging. They met with healthcare providers who initially dismissed the possibility of a dementia diagnosis because the person with dementia was too young or because memory loss was an expected characteristic of older age [20,24,38,44,46,50,54]. Living with pre-diagnosed

**Table 2. Finding numbers related to each synthesized finding and category.**

| Synthesized finding | Category | Finding number (as in Table 1) |
|---|---|---|
| The challenge posed by dementia | Coping | 17–24, 48, 65, 77, 79, 86, 87, 134, 149–153, 181, 208, 242 |
| | Stress and emotions | 17–19, 31–34, 40, 45, 50, 58, 62, 63, 65, 68–72, 74–76, 78–81, 83, 84, 89, 90, 93, 95, 106, 110, 114, 117, 119, 130, 134, 140, 141, 143–146, 148–151, 162, 172, 174–176, 178, 179, 183, 199, 200, 204, 206–214, 217, 233–240, 242 |
| | Finances | 5, 6, 52, 53, 73, 75, 117, 172, 173, 182, 210, 211, 241 |
| | Role/identity | 72, 75, 94, 175, 198, 203, 205, 206, 208, 210, 232 |
| | Noticing changes | 21, 50, 60, 89, 91, 112, 113, 132, 133, 201, 207, 212, 213, 236 |
| | Family support | 5, 10, 25, 36, 39, 40, 43, 44, 46, 48, 49, 52, 55, 59, 82, 86, 114, 137, 154, 170, 171, 178, 186, 191–193, 195–198, 214, 216, 218, 226, 231, 238, 240 |
| | Getting help/adapting | 10, 21, 22, 35, 37, 38, 50, 57, 64, 94, 96, 97, 146, 147, 194, 201, 207, 213, 230, 236, 238 |
| | Communication | 42, 133, 138, 147, 167, 180, 183, 221 |
| | Decision making | 5, 43, 45, 49, 54, 55, 104, 139–141, 143 |
| | Services provided | 16, 41, 42, 44, 45, 47, 49–51, 54, 55, 57–63, 101, 103, 111, 112, 116, 120–126, 128, 131, 135, 137, 141, 143, 145–151, 161, 162, 178, 184, 185, 187, 191, 201, 212, 221, 225, 229, 239, 241 |
| | Information/ understanding | 1–4, 16, 26, 29, 31, 35, 51, 91, 92, 108–112, 116, 132, 151, 163–166, 180, 184, 187, 191, 195, 201, 211, 212, 220–221, 227 |
| Rural limitations | Stress and emotions | 7, 8, 56, 88, 105, 159, 188, 202 |
| | Role/identity | 177 |
| | Family support | 8, 13–15, 28, 85, 98–100, 173, 188, 215 |
| | Getting help/adapting | 12, 13, 98, 99, 168, 188 |
| | Services provided | 9, 27, 28, 67, 88, 100, 102, 105, 107, 115, 118, 136, 159, 160, 169, 223, 224, 226, 228 |
| | Information/ understanding | 7, 28, 30, 67, 88, 159, 160, 169, 202, 243 |
| Rural opportunities | Coping | 66 |
| | Stress and emotions | 127, 155, 156 |
| | Family support | 7, 11, 13, 66, 157, 158, 177, 222, 244, 245 |
| | Getting help/adapting | 11, 13, 66, 142, 158, 189, 190, 244, 245 |
| | Communication | 142 |
| | Services provided | 127, 128 |

dementia was difficult for caregivers [25,29,38,40,54] and caregivers described the need for improved dementia information and education [20,23–26,29,32,38–40,42,46,48,50,51]. They also needed improved communication with and among healthcare providers to be better informed about dementia and the future course of the illness [28,40,42,46,50].

Second, unpaid family caregivers found formal healthcare services (e.g., home care) to be lacking, inflexible and/or uncoordinated [23,29,30,35,39–42,44–46,48,50,51]. As a result, caregivers were left to fill gaps in services by providing services themselves and/or by developing personal networks to help provide care [22,28,34–36,42,44,47]. To alleviate this burden, caregivers wished to have someone who could guide the process of facilitating connections among and across services and providers [40,46,51], especially when providing care became too burdensome for the caregiver to continue safely [41,42,44,48–50].

Third, unpaid family caregivers experienced negative emotions (e.g., sadness, anger) as a part of their caregiving and this was a source of stress [23,25,27–30,32,33,37–41,41,42,44–

46,48–52,55]. While many caregivers found ways to cope with, or adapt to, these negative emotions by seeking support or taking time for self-care activities, participants often reported challenges with finding the support to alleviate their responsibilities and enable them to practice self care [23,28,30,32,42,43,46,49,52,55].

## Synthesized finding 2: Rural limitations

"Rural limitations" describes the barriers or limitations experienced by unpaid family caregivers for persons living with dementia when providing care in rural settings. Six categories contributed to this synthesized finding (stress and emotions, role/identity of caregiver, family support, getting help and/or adapting, services provided, information and understanding).

First, several studies reported that caregivers of persons living with dementia in rural communities felt stigmatized in their community and that the challenges they experienced were not understood by community members [20,22,25,44,45,52,54]. A lack of community member education about dementia (e.g., defining dementia, understanding dementia symptoms, understanding the needs of caregivers) was a challenge which precipitated negative emotions for caregivers [24,25,29,45,54]. Where formal community-based programming for persons living with dementia and their caregivers existed (e.g., day programs, support groups), caregivers noted that they perceived the programming to be more suitable for women than for men (i.e., for both caregivers and persons living with dementia) [34,50].

Second, unpaid family caregivers reported that rural communities had changed due to population aging and the departure of family members for urban areas. As rural communities changed over time, caregivers hesitated to ask for support from remaining community members who had their own health concerns and limitations, as fewer family members were available to provide them with support in rural settings [32,35,43,47,55]. Some caregivers felt that governments did not have the will to provide better supports to help older people age in place in rural areas [37].

Third, unpaid family caregivers felt that the quality of health services in rural communities was inferior to that provided in urban communities. Distance to specialized services led to transportation challenges and excessive time away from home to travel long distances [32,39,40,51]. Where services were provided in their rural communities, caregivers noted a lack of specialized knowledge about dementia (e.g., by family physicians, respite workers, and home care workers who lacked dementia training) [31,48,55] or that, even when a provider had some dementia training, they may not have knowledge appropriate to the specific community and/or culture. For example, participants noted a lack of Indigenous workers in communities with large Indigenous populations [24,51]. As well, rural communities experienced a lack of and/or high turnover in home care and respite workers resulting in inflexible care schedules, inconsistent care providers, and a lack of respite for caregivers [29,30,39,42,44,51,55].

## Synthesized finding 3: Rural opportunities

Study participants often described the benefits and opportunities available for persons living with dementia in rural communities and their unpaid family caregivers. Seven categories contributed to this synthesized finding (coping, stress and emotions, family support, getting help and/or adapting, communication, services provided, information and understanding).

Participants most often cited the strong personal networks in close-knit rural communities and how these affected them positively when caregiving for someone with dementia. These networks are not capable of providing specialized dementia services, but they involve trusting relationships and feelings of mutual responsibility based on the specific local environment. At

times, rural environments were described as positive because of the beauty of the landscape [22], yet positive comments were more often associated with caregivers' feelings of trusting others, being known by others in their close-knit communities, and having community members help supervise persons living with dementia in the community [22,27,31,45,53].

Even where dementia-specific care might not be ideal because of distance from services and local challenges [22], caregivers described the rural setting as a place where they have ties to others, even when rural populations are shrinking [22,45,53]. As a result, participants felt that rural communities provided them with a sense of safety and opportunity when caring for persons living with dementia. A rural network provided a safety net when there was a personal connection to someone who had healthcare knowledge (e.g., a retired nurse), leading caregivers to feel they had someone whom they could call on in a crisis [47].

Family caregivers also felt that rural settings provided opportunities for persons living with dementia. Persons living with dementia were able to link with people and places to participate in activities that provided continuity with their lives before diagnoses (e.g., repairing mechanical parts, gardening, being outdoors with others) [28,31,34]. This was reported as enjoyable and meaningful for persons living with dementia and was beneficial for caregivers because it provided a form of respite for caregivers.

## Discussion

Given the growing incidence of dementia in rural and urban settings, it is important to understand how rurality affects the experiences and needs of caregivers of persons living with dementia in rural communities u. This systematic review of qualitative research uniquely summarizes the experiences and needs of rural family caregivers of persons living with dementia. A meta-aggregate approach was used to establish the synthesized results via coding and categorizing each finding from all included studies and then synthesizing the categories.

These synthesized findings provide new knowledge about unpaid family caregivers' perspectives relating to the limitations and opportunities of living in rural areas while providing care for persons living with dementia. The findings reveal the challenges posed by dementia from the perspectives of caregivers as well as ways in which rural settings may pose limitations and/or provide opportunities when caring for community-dwelling persons living with dementia.

Of note is the importance of relationships to unpaid family caregivers in rural settings. On one hand, caregivers described this setting as limiting due to experiences of stigma yet, on the other hand, the rural setting also benefited caregivers when they experienced trustworthy and helpful relationships in close-knit communities. The importance of addressing stigma in rural settings is documented by others [56,57]. This present review both supports and enriches these findings by demonstrating that unpaid family caregivers may experience stigma in rural communities while, at the same time, often appreciate and rely on close-knit rural relationships. These important relationships are built on small social networks that can help to foster feelings of comfort and safety in rural areas. They also provide crucial opportunities for persons living with dementia to participate in meaningful activities while being supervised by others, thus providing a form of informal respite for unpaid family caregivers when formal respite services may not be available. This is significant in that trustworthy and helpful relationships in rural communities may be protective for unpaid family caregivers. While rural living contributes to challenges for caregivers, this review also captures the experience of rural living as an asset due to the existing social networks in these settings.

This review has implications for health systems and organizations. If unpaid family caregivers experience their communities to be potential assets in the provision of dementia care,

health systems and health organizations could engage and empower existing social networks in rural communities to reduce dementia-related stigma and to co-produce dementia care with community partners. This co-production of care may be facilitated by integrated community care, an approach to care which values the role of the informal care sector, including family caregivers [58]. Intersectoral partners have a role in linking dementia education to communities, increasing access to technology in rural areas through better infrastructure, increasing funding for community-driven programs, and developing local and regional policies to increase community involvement in care.

Implications for research include the need to further investigate both the positive and the negative impacts of rural life on unpaid family caregiving, including identifying outcomes most important to caregivers in rural settings and examining factors that can improve outcomes. Investigating these outcomes and factors will facilitate the development of needed interventions in rural contexts. Methodologically, research using a community-based research approach [59] will build capacity in rural settings for unpaid family caregivers and their communities to direct research projects to best meet their needs and to involve local and regional governments in improving outcomes for caregivers in rural settings.

## Limitations

This review is limited in that the definition of rurality was inconsistent or lacking across studies reviewed, and we cannot know how differing conceptualizations of rurality may have impacted this review. Second, the included studies vary in quality which may have impacted our findings. Lastly, each finding is associated with the experiences and needs of caregivers of persons living with dementia in rural communities, yet we do not directly compare the experiences of rural and urban caregivers in this review. As such, we cannot claim these findings to be unique to rural caregivers, yet our meta-aggregate analysis demonstrates that rural caregivers attribute some of their unmet needs, as well as some of their supports, specifically to their rural contexts.

## Conclusion

This meta-aggregate analysis of qualitative studies describes the experience and needs of rural unpaid family caregivers for persons living with dementia. We have identified three synthesized findings. The first synthesized finding describes the challenges posed by dementia for caregivers, the second describes how caregiving is limited in rural settings, and the third describes the opportunities experienced by family caregivers due to their rural settings. We suggest healthcare systems build on the strengths of rural areas and work with communities to provide dementia care. We also call for future research to utilize community-based approaches to identify outcomes important to rural caregivers.

## Supporting information

**S1 Checklist. PRISMA 2009 checklist.**
(DOC)

## Acknowledgments

Victor Froese, PhD, Library Director at Canadian Mennonite University, aided in the search strategy design for this review.

## Author Contributions

**Conceptualization:** Heather J. Campbell-Enns, Stephen Bornstein, Veronica M. M. Hutchings, Kelli O'Brien, Tara Stewart, Benjamin Rich Zendel, Malcolm B. Doupe.

**Data curation:** Heather J. Campbell-Enns, Maya Janzen, Melanie Kampen.

**Formal analysis:** Heather J. Campbell-Enns, Maya Janzen, Melanie Kampen.

**Funding acquisition:** Heather J. Campbell-Enns, Stephen Bornstein, Veronica M. M. Hutchings, Kelli O'Brien, Tara Stewart, Benjamin Rich Zendel, Malcolm B. Doupe.

**Investigation:** Heather J. Campbell-Enns, Maya Janzen, Melanie Kampen.

**Methodology:** Heather J. Campbell-Enns, Kendra L. Rieger.

**Project administration:** Heather J. Campbell-Enns.

**Supervision:** Heather J. Campbell-Enns.

**Writing – original draft:** Heather J. Campbell-Enns, Stephen Bornstein, Veronica M. M. Hutchings, Kelli O'Brien, Kendra L. Rieger, Tara Stewart, Benjamin Rich Zendel, Malcolm B. Doupe.

**Writing – review & editing:** Heather J. Campbell-Enns, Stephen Bornstein, Veronica M. M. Hutchings, Maya Janzen, Melanie Kampen, Kelli O'Brien, Kendra L. Rieger, Tara Stewart, Benjamin Rich Zendel, Malcolm B. Doupe.

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
