## [Decision Letter · Decision Letter 0]

27 Feb 2023

PONE-D-23-02151The experiences and needs of unpaid family caregivers for persons living with dementia in rural settings: A qualitative systematic reviewPLOS ONE

Dear Dr. Campbell-Enns,

Thank you for submitting your manuscript to PLOS ONE. After careful consideration, we feel that it has merit but does not fully meet PLOS ONE’s publication criteria as it currently stands. Therefore, we invite you to submit a revised version of the manuscript that addresses the points raised during the review process. Please substantially revise your article in light of the comments of all three reviewers.

We look forward to receiving your revised manuscript.

Kind regards,

Anastassia Zabrodskaja, Ph.D.

Academic Editor

PLOS ONE

Journal Requirements:

"The authors have declared that no completing interests exist."

Reviewers' comments:

Reviewer's Responses to Questions

**Comments to the Author**

1. Is the manuscript technically sound, and do the data support the conclusions?

Reviewer #1: Yes

Reviewer #2: No

Reviewer #3: Partly

2. Has the statistical analysis been performed appropriately and rigorously? 

Reviewer #1: N/A

Reviewer #2: No

Reviewer #3: N/A

3. Have the authors made all data underlying the findings in their manuscript fully available?

Reviewer #1: Yes

Reviewer #2: No

Reviewer #3: Yes

4. Is the manuscript presented in an intelligible fashion and written in standard English?

Reviewer #1: Yes

Reviewer #2: No

Reviewer #3: Yes

5. Review Comments to the Author

Reviewer #1: The authors have addressed an important area of concern n the care of people with dementia in relation to place of residence, and the article is based on robust data with global spread. The only snag is the paucity of papers from LMICs which abound with rural settings for comparison based on socio-economic standings.

The only concern is the stereotyping with the use of "rural persons, rural caregivers of rural families" as found on lines 256, 284, 313, 358-360. These should be revised to read " caregivers of persons/families resident in rural communities"

Reviewer #2: This paper needs to revise in every section. Need to write the paper according to the template of the journal. Please check the organization of the paper. The updated references should be added. A few of the research directions should be implemented.

Reviewer #3: • In paper, a cohort of studies was selected as there is only one study published in 1993 which is mentioned in Table (1), while the remaining 35 studies are from 2000-2022. How was this cohort decided? If researchers intended to add research published in 2000-22 then they should exclude 1993.

• In paper, study rationale and implication are missing. Add some paragraphs about study implications and significance.

• Researchers selected studies from Canada (n=15), the 187 United States (n=7), Australia (n=3), Scotland (n=3), Norway (n=1), Sweden (n=1), South Africa 188 (n=1), Kenya (n=1), Ghana (n=1), and Japan (n=1). As Japan is an Asian country then researches from other Asian countries like Pakistan, India, Sri Lanka, China, Iran, Malaysia, were not included? kindly write down the reasons behind selection of countries.

• Researchers just gave overall findings of researches, at least they should specify the category of caregiver, no doubt, in method section, they have described caregiver” was defined as an unpaid career who may be referred to as a care partner or informal career; But in Table 1, a separate column should be made for the extracted themes to elaborate that family caretaker might have different concerns than those informal careers.

• Selected studies did not specify that patients with dementia were suffering from other morbidities or not, and also did not explain that patients might be completely paralyzed or partially paralyzed or were quite stable if in the early stage of dementia.

• Studies did not specify the age of caregivers and gender. Age and gender also bring different responses.

• In discussion, there is a need to describe findings from formal (family members) and informal careers separately.

6. PLOS authors have the option to publish the peer review history of their article (what does this mean?). If published, this will include your full peer review and any attached files.

Reviewer #1: **Yes: **ADESOLA OGUNNIYI

Reviewer #2: No

Reviewer #3: **Yes: **HAMEDI M.A.

---

## [Author Response · Author response to Decision Letter 0]

5 May 2023

Reviewer 1: Thank you for your suggestions and comments. We have incorporated your suggestions into the revision. 

Reviewer 2: Thank you for your comments and suggestions. We have revised for all suggestions. 

Reviewer 3: Thank you for your comments and suggestions. We have incorporated all your suggestions into the revision and added further comments to clarify in the letter of response to the reviewers.

---

## [Editor Report · Decision Letter 1]

18 May 2023

The experiences and needs of unpaid family caregivers for persons living with dementia in rural settings: A qualitative systematic review

PONE-D-23-02151R1

Dear Dr. Campbell-Enns,

We’re pleased to inform you that your manuscript has been judged scientifically suitable for publication and will be formally accepted for publication once it meets all outstanding technical requirements.

Kind regards,

Anastassia Zabrodskaja, Ph.D.

Academic Editor

PLOS ONE
---

## [Editor Report · Acceptance letter]

6 Jun 2023

PONE-D-23-02151R1 

The experiences and needs of unpaid family caregivers for persons living with dementia in rural settings: A qualitative systematic review 

Dear Dr. Campbell-Enns:

I'm pleased to inform you that your manuscript has been deemed suitable for publication in PLOS ONE. Congratulations! Your manuscript is now with our production department. 

Kind regards, 

on behalf of

Professor Anastassia Zabrodskaja 

Academic Editor

PLOS ONE